# Kynurenine/Tryptophan Ratio Predicts Angiotensin Receptor Blocker Responsiveness in Patients with Diabetic Kidney Disease

**DOI:** 10.3390/diagnostics10040207

**Published:** 2020-04-09

**Authors:** Ming-Hsien Wu, Chia-Ni Lin, Daniel Tsun-Yee Chiu, Szu-Tah Chen

**Affiliations:** 1Division of Endocrinology and Metabolism, Department of Internal Medicine, Chang Gung Memorial Hospital, Chang Gung University, College of Medicine, Taoyuan 333, Taiwan; b9502013@cgmh.org.tw; 2Division of Endocrinology and Metabolism, Department of Internal Medicine, New Taipei Municipal TuCheng Hospital, New Taipei City 236, Taiwan; 3Department of Laboratory Medicine, Chang Gung Memorial Hospital Linkou Branch, Chang Gung University, Taoyuan 333, Taiwan; chianilin@cgmh.org.tw; 4Department of Medical Biotechnology and Laboratory Science, College of Medicine, Chang Gung University, Taoyuan 333, Taiwan; dtychiu@mail.cgu.edu.tw; 5Healthy Aging Research Center, Chang Gung University, Taoyuan, 333, Taiwan

**Keywords:** macroalbuminuria, microalbuminuria, kynurenine, tryptophan, angiotensin receptor blocker

## Abstract

Albuminuria is a measurement and determinant factor for diabetic kidney disease (DKD). Angiotensin receptor blocker (ARB) is recommended for albuminuria in DKD with variable response. To find surrogate markers to predict the therapeutic effect of ARB, we carried out a prospective study to correlate plasma metabolites and the progression of renal function/albuminuria in DKD patients. A total of 56 type 2 diabetic patients with various stages of chronic kidney disease and albuminuria were recruited. ARB was prescribed once albuminuria was established. Urinary albumin-to-creatinine ratio (UACR) was determined before and six months after ARB treatment, with a ≥30% reduction of UACR considered an ARB responder. Plasma levels of 145 metabolites were measured before ARB treatment; only those associated with albuminuria were selected and compared between ARB responders and non-responders. Both lower tryptophan (Trp ≤ 46.75 μmol/L) levels and a higher kynurenine/tryptophan ratio (KTR ≥ 68.5 × 10^−3^) were significantly associated with macroalbuminuria (MAU), but only KTR (≥54.7 × 10^−3^) predicts ARB responsiveness (sensitivity 90.0%, specificity 50%) in MAU. Together, these data suggest that the kynurenine/tryptophan ratio predicts angiotensin receptor blocker responsiveness in patients with diabetic kidney disease.

## 1. Introduction

Diabetic kidney disease (DKD) is defined as a decline in renal filtration and/or the development of albuminuria, which occurs in approximately 40% of diabetic patients and is the leading cause of chronic kidney disease (CKD) worldwide [1]. The natural course of DKD initiates with glomerular hyperfiltration, and goes downhill with progressive albuminuria, declining glomerular filtration rate (GFR), and ultimately ends up with end-stage renal disease (ESRD) if inappropriately treated [2]. Albuminuria accelerates kidney disease progression through multiple mechanisms including the induction of tubular chemokine and complement reaction, interstitial inflammatory cell infiltration, and sustained fibrogenesis [3,4,5]. The extent of albuminuria is generally regarded as a marker of the severity of CKD and the estimation of the future decline in GFR at clinical practice [6].

Many randomized control trials in recent decades have proven that the reduction of albuminuria may slow the progression of nephropathy [4,6,7,8]. Both angiotensin-converting enzyme inhibitors (ACEi) and angiotensin receptor blockers (ARB) are antihypertensive drugs acting through the inhibition of the renin-angiotensin-aldosterone system (RAAS). Meanwhile, ACEi/ARB are effective in the protection of glomerular endothelium through a reduction in intraglomerular pressure and urinary albumin excretion [9].

Previous metabolomics studies have revealed potential pathways and surrogate prognostic markers associated with the development and progression of DKD [10,11]. Nevertheless, little is known about the metabolic change during the progression of albuminuria among DKD patients. Thus, we profiled plasma acylcarnitines, amino acids, biogenic amines and sphinogolipids/glycerophospholipids by targeted metabolomics in type 2 diabetes mellitus (T2DM) patients with or without albuminuria in this work. The aim of our study is to investigate the changes in metabolites in various stages of CKD, and to verify the predictive value of these metabolites on the progression of DKD and the effectiveness of ARB treatment.

## 2. Materials and Methods

### 2.1. Patients

With informed consent approved by the Ethics Committee on Research of the institutional review board (approval number: 201600692B0, 3 August 2016), T2DM patients aged from 35 to 75 years, regularly followed-up at the out-patient department of a tertiary hospital, were screened. Patients with acute inflammatory diseases, severe systemic disease, and/or ACEi/ARB vulnerability were excluded. Eventually, 56 patients with various stages of CKD with (*n* = 48) or without (*n* = 8) albuminuria were enrolled from April 2017 to May 2018. The presence of albuminuria was assessed by at least two measurements of the urinary albumin-to-creatinine ratio in a random spot urine collection. While normoalbuminuria means a UACR < 30 mg/g, microalbuminuria and macroalbuminuria are defined as when UACR ranges between 30–300 mg/g and UACR ≥ 300 mg/g, respectively [12,13]. Once albuminuria was established, all patients were judiciously treated with ARB according to their blood pressure levels. For metabolite measurement, plasma samples were collected at the diagnosis of albuminuria in ARB naive patients or collected 4 weeks after a drug holiday for ACEi/ARB. Patients with more than a 30% decrease in the amount of UACR were defined as responders, according to previous reports [14]. A total of 34 macroalbuminuria (MAU) and 14 microalbuminuria (mau) patients were enrolled in this study; finally, 20/34 of the MAU and 7/14 of the mau patients were ARB responders after a 6-month period of follow up.

### 2.2. Metabolomic Approach

Metabolite levels can be regarded as the ultimate response of biological systems to pathological mechanisms. To investigate if metabolomics can be used to identify novel clinical biomarkers and therapeutic targets for DKD, plasma samples were collected from T2DM patients with various degrees of albuminuria, after an overnight fast, for metabolite analysis. Blood sample were collected with defined clinical parameters by diabetologists from the diabetic clinic of a medical center. A targeted quantitative metabolomics approach using a combined liquid chromatography MS/MS assay and direct flow injection assay (AbsoluteIDQTM180 kit from Biocrates Life Science, Innsbruck, Austria) was used for the metabolomics analyses of the samples. The assay was performed in Waters Acquity Xevo TQ-S instrument according to manufacturer’s instruction. The metabolomics dataset contained 20 acylcarnitine, 21 amino acids, 8 biogenic amines, 14 sphingomyelins, and 82 glycerophospholipids.

### 2.3. Statistical Analysis

Continuous variables were presented as mean ± standard deviation (SD) and range, categorical variables were presented as number and percentage. The comparisons of the characteristics were calculated by a one-way ANOVA for continuous variables with normal distribution; a Kruskal–Wallis ANOVA was used for continuous variables without normal distribution, and a Chi-Squared test was used for categorical variables.

An independent sample t-test was used for continuous variables with normal distribution and a Mann–Whitney U-test for continuous variables without normal distribution to analyze the difference between ARB responders and non-responders in both MAU and mau groups. The receiver operating characteristic (ROC) curve and Youden Index were carried out to identify the most predictive value of Trp and KTR for albuminuria and KTR for predicting ARB responsiveness in MAU. The adjusting confounding factors about KTR between responders and non-responders in the MAU group was determined by multivariate binary logistic regression. Analysis was performed using SPSS statistical software (version 22.0, SPSS Inc., Chicago, IL, USA). A *p* value < 0.05 was considered statistically significant.

## 3. Results

Table 1 summarizes the demographic characteristics of 56 (30 male, 26 female) T2DM patients with various degrees of albuminuria. Patients were divided into three groups including macroalbuminuria (MAU, *n* = 34), microalbuminuria (mau, *n* = 14), and normoalbuminuria (control, *n* = 8) according to their initial urinary albumin-to-creatinine ratio (UACR) before the enrollment. Despite that patients with MAU were significantly younger, the other parameters, such as body mass index (BMI), glycated hemoglobin (HbA1c), duration of diabetes, and blood pressure before ARB treatment were comparable among the three groups. Meanwhile, the use of oral anti-diabetic drugs and anti-hypertensive drugs were all similar among the three groups.

The average UACR was 1831.0 ± 1640.5 in MAU, 146.5 ± 85.9 in mau, and 10.6 ± 6.4 mg/g in control groups, respectively, with statistical significance (*p* < 0.001). The average baseline estimated glomerular filtration rate (eGFR) was similar (42.9 ± 18.6, 50.1 ± 12.8, and 49.2 ± 12.8 mL/min/1.73 m^2^, *p* = 0.325) among the three groups, with 11.8%, 64.7%, 23.5%; 28.6%, 64.3%, 7.1% and 12.5%, 87.5%, 0% of the patients allocated in CKD stage II, III, and IV–V, respectively.

Plasma collected from the control and albuminuria patients before ARB treatment were analyzed by an AbsoluteIDQTM180 kit to investigate the metabolomic change. Out of the 145 metabolites examined, nine metabolites including five amino acids, i.e., serine (Ser), tryptophan (Trp), tyrosine (Tyr), ornithine (Orn), phenylalanine (Phe); 3 glycerophospholipid, PC ae C44:3, lysoPC a C24:0, lysoPC a C26:1; and 1 sphingolipid, SM C26:0, showed significant differences among the three groups (Table 2). In contrast to the other metabolites showing a negative association with the severity of albuminuria, the plasma levels of Orn and Phe were higher in the albuminuric, especially in the mau group. In addition, although the plasma levels of the biogenic amine kynurenine (Kyn, a degradation product of Trp) was similar among the three groups, the kynurenine/tryptophan ratio (KTR) was significantly higher in the MAU group (*r* = 0.33, *p* = 0.013, Figure 1A) probably due to the negative correlation between Trp and the severity of albuminuria (*r* = −0.32, *p* = 0.017, Figure 1B).

Since that a correlation between plasma Trp and rapid progression of DKD has been reported [11], we tried to verify whether low Trp and/or high TKR in patients with albuminuria may also predict the progression of DKD and the responsiveness of ARB in a group of patients with similar stages of CKD but various degrees of albuminuria in this study. Thus, UACR were evaluated six months after ARB treatment. An ARB responder was arbitrarily defined as an over 30% reduction of UACR. As a consequence, 20 out of 34 MAU and 7 out of 14 mau patients were classified as responders six months after ARB treatment. When the nine aforementioned metabolites associated with albuminuria were analyzed, none of the metabolites, except KTR, showed a significant difference between responders and non-responders regardless in the MAU or mau groups. As yet, it was interestingly to find that even though neither plasma level of Trp nor Kyn was changed by ARB, the ratio of the metabolite and its degradation product, i.e., KTR, revealed significant association (*p* = 0.025) to predict ARB responsiveness in the MAU, but not in the mau (*p* = 0.085, Table 3) patients. Furthermore, when the confounding factors, such as age, SBP, eGFR, gender, HbA1c, duration of DM, and the use of dipeptidyl-peptidase-4 (DPP4) inhibitor, glucagon-like peptide-1 receptor (GLP-1R) agonist or sodium glucose co-transporters 2 (SGLT2) inhibitor were taken into consideration, KTR remained significant difference between responders and non-responders in the MAU group (Table 4).

To detect the optimal cut-off point of Trp and KTR in predicting the development of macroalbuminuria and the effectiveness of ARB treatment, receiver operating characteristic curve analysis and the Youden Index were applied, with a cut-off value of Trp ≤ 46.75 μmol/L (sensitivity 61.8%, specificity 77.3%) and KTR ≥ 68.5 × 10^−3^ (sensitivity 50.0%, specificity 86.4%) for albuminuria, and KTR ≥ 54.7 × 10^−3^ (sensitivity 90.0%, specificity 50%) for ARB responsiveness, respectively (Figure 2).

## 4. Discussion

The clinical diagnosis of DKD is on the basis of a patient with long-standing diabetes with albuminuria and/or reduced eGFR in the absence of signs or symptoms of other primary causes of kidney damage [15]. Lines of evidence have proven that albuminuria is strongly associated with the progression of CKD in both diabetic and non-diabetic patients [16,17]. Chronic proteinuria is attributed to a loss of selectivity of the glomerular barrier to protein filtration and further compels renal tubules to carry on excessive resorption. In addition, protein overload induces the release of cytokines, chemokines, vasoactive molecules, and inflammatory proteins from tubular cells to trigger apoptosis, monocyte infiltration in the interstitium of extracellular matrix, and eventually tissue scarring and GFR loss [18,19]. Thus, the finding of a surrogate clinical marker that may precisely predict the bane of albuminuria may benefit in preventing or delaying the progression of DKD.

To reach this goal, we correlated the change of metabolites in T2DM patients with various stages of CKD in this study, and we found a trend of decreased metabolites (Ser, Trp, Tyr, PC ae C44:3, lysoPC a C24:0, lysoPC a C26:1, SM C26:0), except Orn and Phe, showing significant association with the severity of albuminuria. Several amino acids, including Orn, were reported to be positively correlated with UACR [20]. Here, we also found higher Orn levels in the albuminuric, especially in the mau, rather than in the MAU group of patients. Although we cannot fully explain the discrepancy, one of the possible reasons may be attributed to the use of SGLT-2 inhibitors in the MAU group. Increased ornithine decarboxylase (ODC) activity may lead to hyperfiltration in a diabetic kidney [21], and any factors interfering eGFR may influence the activity of ODC and further modulate the concentration of Orn. For example, in our cases, SGLUT-2 inhibitors may decrease eGFR due to their promotion of glomerular afferent arteriole constriction [22].

Plasma levels of Phe are regulated by phenylalanine hydroxylase (PAH); factors such as stress, sepsis, or any causes of hepatic dysfunction may influence the activity of PAH [23,24]. In accordance with studies reporting increased plasma Phe levels in CKD patients with reduced renal PAH activity [25], we also found higher Phe levels in albuminuric patients, albeit we cannot explain why the Phe level was higher in the mau rather than in the MAU patients. Corresponding to the previous studies [11,26], we also found a significant inverse correlation between the plasma levels of Trp (cut-off value 46.75 μmol/L, sensitivity 61.8%, specificity 77.3%) and the severity of albuminuria. The essential amino acid Trp is indispensable for the synthesis of many biomolecules, such as enzymes, structural proteins, serotonin, and neurotransmitters important to sustain normal life. The majority (~95%) of free Trp undergoes oxidative metabolism along the Kyn pathway by using the tryptophan 2,3-dioxygenase (TDO) and indoleamine 2,3-dioxygenase (IDO). TDO exists only in the liver, whereas IDO exists ubiquitously in the body (Figure 3). Under normal physiological conditions, TDO controls > 95% of Trp degradation in the liver and hence has a higher capacity for the regulation of Trp metabolism through the Kyn pathway in comparison to IDO [27].

However, IDO is potently induced by proinflammatory cytokines and acts locally to modulate Trp levels in response to inflammation [26,28]. Therefore, the ratio of Trp and its degradation product, Kyn; i.e., KTR, is frequently utilized to evaluate the activity of the extrahepatic Trp degrading enzyme, IDO [29]. Notably, many determinants can influence the KTR value, for example, each of the four main pathways of Trp metabolism including serotonin, tryptamine, indolepyruvic acid, and the Kyn pathway can disturb the ratio [30,31]. Although many cytokines and their mediators (both proinflammatory and anti-inflammatory) may also affect IDO, IFN-γ is still the principal effector of IDO activity [32,33]. In the situation of diabetes, elements of the diabetic environment can directly or indirectly activate T cells in diabetic kidneys. CD4^+^ T cells produce IFN-γ in order to respond to advanced glycation end products and thus exacerbate inflammation in the diabetic kidney [34]. Correspondingly, our data also showed a significant positive correlation between KTR (cut-off value 68.5 × 10^−3^) and the severity of albuminuria. Similarly, Zhao et al. also reported that plasma KTR could be applied as a reliable biomarker for the assessment of renal function in patients with hypertension [35].

The renin–angiotensin system blockages are valid of renoprotection beyond blood pressure control [36]. Although the mechanisms involving renoprotection of ARB are complex, it is generally believed that a direct vasodilation of the afferent arterioles may contribute to an increase in renal blood flow, and lead to an improvement of renal ischemia and hypoxia. In addition, ARB reduces intraglomerular pressure and protects glomerular endothelium and/or podocyte injuries. In addition, ARB blocks local angiotensin II-induced renal cell and tissue injuries [9]. Thus, ACEi/ARB was suggested as the standard practice to reduce the risk of DKD in T2DM patients [37,38,39]. The progression of macroalbuminuria was considered irreversible and to be the “point of no return” in the past [7,40,41]; however, with intensive treatment, Hiroki et al. reported that remission of macroalbuminuria occurs frequently in T2DM patients [42], and indeed, delayed progression from macroalbuminuria toward end-stage renal disease was assured in the IDNT and the RENAAL trials with the application of ARBs [43,44].

As our treatment protocol, ARB was prescribed and UACR was followed-up six months after treatment. Among the 48 patients with albuminuria, we found that 56.3% (27/48) of these patients had more than 30% UACR reduction. Among the 27 responders, 20 were initially MAU and seven were mau; in addition, five of the MAU patients regressed from macroalbuminuria to microalbuminuria after ARB treatment. Subsequently, the aforementioned metabolites associated with albuminuria were analyzed and only KTR was found to be associated with ARB responsiveness, albeit only in the MAU, but not in the mau group. A higher level of KTR (≥ 54.7 × 10^−3^) might be regarded as a potential prognostic marker for improvement of albuminuria, especially in patients with macroalbuminuria. Impressively, although the MAU group was significantly younger, KTR remained at a significant difference after adjusting for confounding factors such as age, gender, SBP, eGFR, HbA1c, duration of DM, and the use of SGLT2 inhibitor (Table 4).

The regulatory role of angiotensin II and AT1 receptor on the Trp/Kyn pathway remains unknown. Although Zakrocka et al. reported that ARB directly inhibits Kyn aminotransferases activity (Figure 3) in the rat brain [45], it remains unclear whether this may also occur in the other organs, such as the kidney. As an assumption, we infer the difference depending on the situation of macroalbuminuria may reflect a more severe glomerular damage with tubulointerstitial fibrosis but microalbuminuria is not [46]. Thus, our work showing significantly higher KTR in MAU patients may indicate that patients with more severe or ongoing inflammatory process may benefit more from ARB treatment.

Our study had some limitations. First, a relatively small number of patients with microalbuminuria were included. Second, the period of ARB treatment was too short to monitor the long term change of renal function. Third, plasma concentrations of indoxyl sulfate and oxidative metabolites degraded from Trp as well as the degradation product of Kyn, i.e., kynurenic acid, were not measured in this study. Fourth, we did not have any information about the KTR level after treatment of ARB. Last, as an essential amino acid, the daily intake of Trp was not fully quantified in this study since that bias might exist under extremely high or low protein intake.

## 5. Conclusions

Despite the mechanism remaining obscured, extensive albuminuria may be associated with lower levels of plasma Trp (≤46.75 μmol/L) and a higher ratio of kynurenine versus tryptophan (KTR ≥ 68.5 × 10^−3^). Meanwhile, a higher KTR ≥ 54.7 × 10^−3^ may predict ARB responsiveness in the MAU group. We conclude that tryptophan and/or kynurenine could play an important role in diabetic nephropathy and can be used to evaluate the response of ARB treatment in macroalbuminuria patients.

## Figures and Tables

**Figure 1 diagnostics-10-00207-f001:**
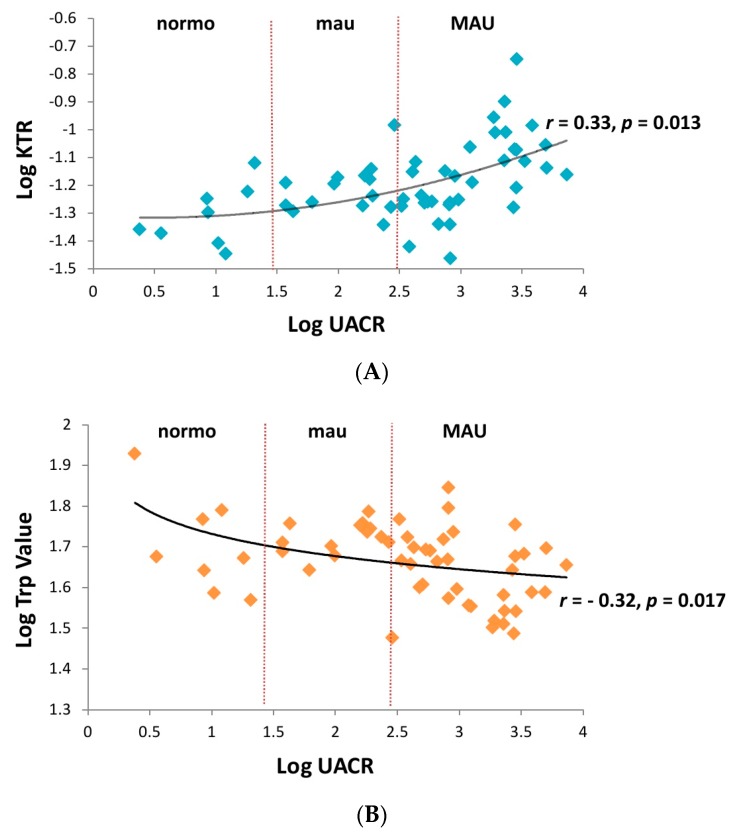
(**A**) The serum kynurenine/tryptophan ratio (KTR) revealed a significantly positive correlation with the severity of albuminuria (*r* = 0.33, *p* = 0.013). (**B**) Serum trptophan (Trp) showed a significantly negative correlation with the severity of albuminuria (*r* = −0.32, *p* = 0.017). MAU, macroalbuminuria; mau, microalbuminuria; normo, normoalbuminuria.

**Figure 2 diagnostics-10-00207-f002:**
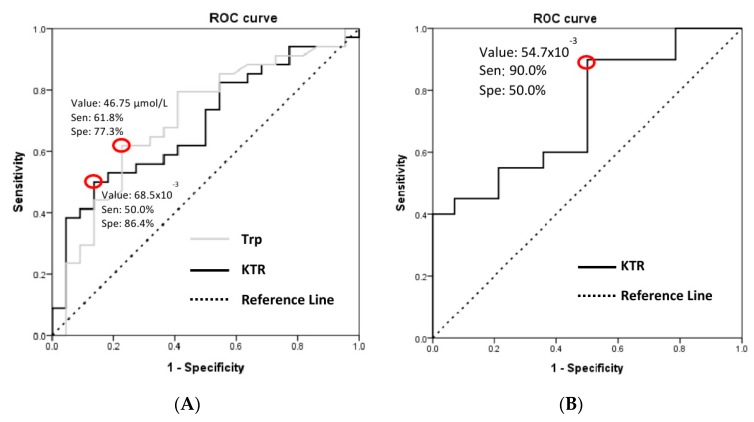
ROC curve analysis was carried out by SPSS (SPSS, Chicago, IL, USA) to determine the best discrimination point of (**A**) Trp (cut-off value 46.75 μmol/L with sensitivity 61.8% and specificity 77.3%, AUC = 0.704 with 95% CI 0.562–0.846, *p* = 0.011, SD = 0.073) and KTR (cut-off value 68.5 × 10^−3^ with sensitivity 50.0% and specificity 86.4%, AUC = 0.687 with 95% CI 0.547–0.827, *p* = 0.019, SD = 0.071) for predicting MAU. (**B**) KTR for ARB responsiveness in diabetic patients with MAU (cut-off value 54.7 × 10^−3^ with sensitivity 90.0% and specificity 50.0%, AUC = 0.729 with 95% CI 0.558–0.899, *p* = 0.025, SD = 0.087). ROC, receiver operating characteristic; Trp, tryptophan; KTR, kynurenine to tryptophan ratio; ARB, angiotensin receptor blocker; AUC, area under the ROC curve; 95% CI, 95% confidence interval; SD, standard deviation.

**Figure 3 diagnostics-10-00207-f003:**
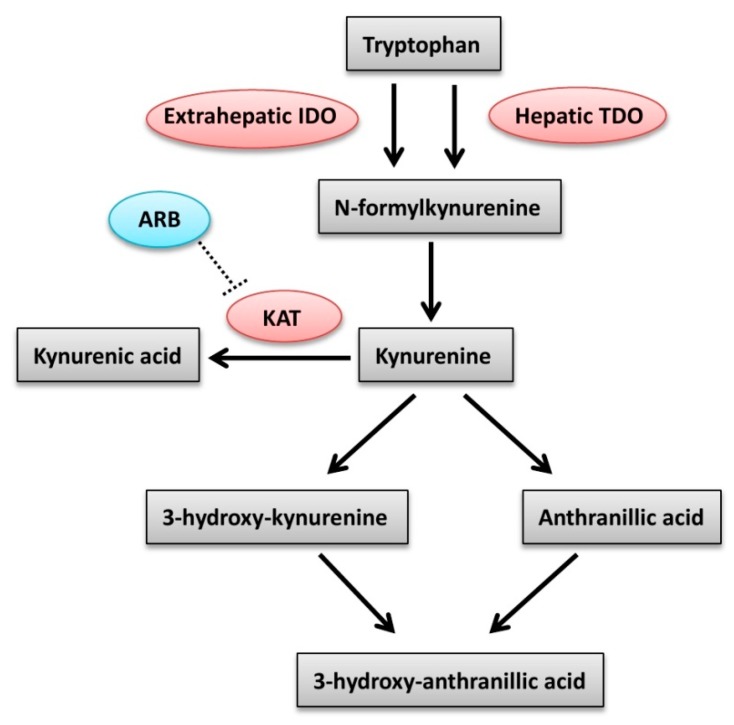
The kynurenine pathway of tryptophan degradation in mammals. IDO, indoleamine 2,3-dioxygenase; TDO, tryptophan 2,3-dioxygenase; ARB, angiotensin receptor blocker; KAT, kynurenine aminotransferase.

**Table 1 diagnostics-10-00207-t001:** Characteristics of 56 patients with various levels of albuminuria.

Variable	MAU (*n* = 34)	Mau (*n* = 14)	Control (*n* = 8)	*p* Value
Gender				
Male	20 (58.8)	6 (42.9)	4 (50.0)	0.597
Female	14 (41.2)	8 (57.1)	4 (50.0)	
Age (years)	61.5 ± 8.5	68.1 ± 5.6	66.9 ± 4.6	0.013 *
BMI (kg/m^2^)	27.9 ± 4.8	26.4 ± 5.2	25.4 ± 3.4	0.341
HbA1c (%)	7.73 ± 1.31	7.29 ± 0.74	7.34 ± 0.44	0.393
Duration of DM (years)	13.9 ± 7.3	13.3 ± 5.7	14.8 ± 6.8	0.890
SBP (mmHg)	134.7 ± 10.9	134.9 ± 8.35	137.1 ± 11.0	0.833
DBP (mmHg)	78.0 ± 11.5	76.3 ± 6.12	77.8 ± 11.6	0.873
UACR (mg/g) ^‡^	1831.0 ± 1640.5	146.5 ± 85.9	10.6 ± 6.4	<0.001 *
Cr (mg/dL) ^‡^	1.92 ± 1.32	1.27 ± 0.26	1.33 ± 0.30	0.103
eGFR (mL/min/1.73 m^2^) ^‡^	42.9 ± 18.6	50.1 ± 12.8	49.2 ± 12.8	0.325
CKD stage II	4 (11.8)	4 (28.6)	1 (12.5)	0.314
CKD stage III	22 (64.7)	9 (64.3)	7 (87.5)	
CKD stage IV, V	8 (23.5)	1 (7.1)	0	
OAD with				
Metformin	14 (41.2)	7 (50.0)	6 (75.0)	0.260
Sulfonylurea	24 (70.6)	10 (71.4)	6 (75.0)	1.000
DPP4 inhibitor	15 (44.1)	9 (64.3)	5 (62.5)	0.412
GLP-1R agonist SGLT2 inhibitor	4 (11.2) 5 (14.7)	0 0	0 0	0.332 0.308
Insulin injection	13 (38.2)	3 (21.4)	2 (25.0)	0.603
Anti-hypertensive drugs				
Beta-blacker	8 (23.5)	7 (50.0)	2 (25.0)	0.195
CCB	22 (64.7)	10 (71.4)	4 (50.0)	0.569
Diuretics	4 (11.8)	3 (21.4)	0	0.376

Continuous variables presented as mean ± standard deviation or *n* (%). One-way analysis of variance (ANOVA) was used for continuous variables with normal distribution; Kruskal-Wallis ANOVA was used for continuous variables without normal distribution (superscript **^‡^**). Chi-Squared Test was used for categorical variables. Superscript * means *p* < 0.05. Definition of albuiminuria: UACR ≥ 300 mg/g defines MAU; UACR between 30 and 300 mg/g defines mau; UACR < 30 mg/g defines normoalbuminuria. Abbreviation: MAU, macroalbuminuria; mau, microalbuminuria; normo, normoalbuminuria; BMI, body mass index; HbA1c, glycated hemoglobin A1c; DM, diabetes mellitus; SBP, systolic blood pressure; DBP, diastolic blood pressure; UACR, urine albumin-to-creatinine ratio; Cr, creatinine; eGFR, estimated glomerular filtration rate; CKD, chronic kidney disease; OAD, oral antidiabetic drug; DPP4, dipeptidyl-peptidase-4; GLP-1R, glucagon-like peptide-1 receptor; SGLT-2, sodium glucose co-transporters 2; CCB, calcium channel blocker.

**Table 2 diagnostics-10-00207-t002:** Metabolites showed a significant association with various stages of albuminuria in 56 T2DM patients.

Metabolites	MAU (*n* = 34)	Mau (*n* = 14)	Control (*n* = 8)	*p* Value
Amino acids				
Ser	99.9 ± 25.0	119.9 ± 31.8	126.0 ± 24.8	0.016 *
Trp	44.5 ± 9.32	51.3 ± 7.62	52.3 ± 15.7	0.042 *
Tyr	53.3 ± 10.8	66.1 ± 7.57	64.6 ± 16.7	0.001 *
Orn ^‡^	122.7 ± 35.3	158.1 ± 68.2	95.9 ± 31.8	0.020 *
Phe ^‡^	65.7 ± 13.6	75.1 ± 10.3	66.9 ± 13.2	0.007 *
Biogenic amines				
Kyn	3.10 ± 0.89	3.14 ± 0.56	2.56 ± 0.72	0.207
Kyn/Trp	0.073 ± 0.028	0.062 ± 0.014	0.050 ± 0.013	0.046 *
Glycerophospholipids				
PC ae C44:3 ^‡^	0.102 ± 0.022	0.118 ± 0.028	0.126 ± 0.025	0.025 *
lysoPC a C24:0	0.138 ± 0.032	0.164 ± 0.031	0.166 ± 0.034	0.013 *
lysoPC a C26:1 ^‡^	0.027 ± 0.009	0.034 ± 0.008	0.034 ± 0.008	0.006 *
Sphingolipids				
SM C26:0	0.208 ± 0.047	0.242 ± 0.062	0.249 ± 0.041	0.036 *

By one-way ANOVA and Kruskal–Wallis one-way ANOVA (superscript **^‡^**), nine metabolites and Kyn/Trp showed a significant association with albuminuria change but Kyn did not reach statistical significance. The mean concentration–standard deviation of these metabolites is indicated in each stage of albuminuria. Superscript * means *p* < 0.05. MAU, macroalbuminuria; mau, microalbuminuria; normo, normoalbuminuria; Ser, Serine; Trp, tryptophan; Tyr, tyrosine; Orn, ornithine; Phe, phenylalanine; Arg. arginine; Kyn, kynurenine.

**Table 3 diagnostics-10-00207-t003:** Metabolites associated with angiotensin receptor blocker responsiveness in 48 T2DM patients with various stages of albuminuria.

Metabolites	MAU Group			Metabolites	Mau Group		
	Responder (*n* = 20)	Non-Responder (*n* = 14)	*p* Value		Responder (*n* = 7)	Non-Responder (*n* = 7)	*p* Value
Amino acids				Amino acids			
Ser	95.7 ± 25.7	105.1 ± 24.1	0.306	Ser	129.3 ± 41.3	110.4 ± 16.5	0.296
Trp	42.4 ± 6.84	47.6 ± 11.6	0.108	Trp ^‡^	52.2 ± 4.75	50.3 ± 10.1	0.749
Tyr	53.7 ± 10.3	52.8 ± 11.9	0.805	Tyr	66.1 ± 5.10	66.1 ± 9.91	1.000
Orn ^‡^	126.3 ± 29.7	118.3 ± 41.9	0.351	Orn	165.6 ± 57.2	150.6 ± 81.6	0.698
Phe ^‡^	68.4 ± 15.6	61.8 ± 9.10	0.178	Phe ^‡^	78.3 ± 11.6	71.9 ± 8.51	0.142
Biogenic amines				Biogenic amines			
Kyn ^‡^	3.32 ± 0.97	2.79 ± 0.67	0.112	Kyn	2.93 ± 0.59	3.34 ± 0.48	0.175
Kyn/Trp ^‡^	0.081 ± 0.031	0.060 ± 0.015	0.025 *	Kyn/Trp ^‡^	0.056 ± 0.008	0.069 ± 0.017	0.085
Glycerophospholipids				Glycerophospholipids			
PC ae C44:3 ^‡^	0.103 ± 0.025	0.100 ± 0.018	0.834	PC ae C44:3	0.116 ± 0.036	0.121 ± 0.020	0.746
lysoPC a C24:0	0.135 ± 0.028	0.141 ± 0.037	0.608	lysoPC a C24:0	0.153 ± 0.029	0.175 ± 0.030	0.187
lysoPC a C26:1 ^‡^	0.027 ± 0.010	0.026 ± 0.006	0.888	lysoPC a C26:1	0.032 ± 0.009	0.036 ± 0.007	0.297
Sphingolipids				Sphingolipids			
SM C26:0	0.209 ± 0.054	0.208 ± 0.038	0.969	SM C26:0	0.254 ± 0.079	0.231 ± 0.040	0.494

In order to evaluate the predictive value of the indicated metabolites in albuminuria response to ARB responsiveness, the serum concentrations of nine metabolites were compared between patients with an UACR change ≥30% six months after ARB treatment to those <30% by an independent sample t-test for continuous variables with normal distribution and a Mann–Whitney U-test for continuous variables without normal distribution (superscript **^‡^**). The kynurenine to tryptophan ratio showed a significant association (*p* = 0.025). Superscript * means *p* < 0.05. Ser, Serine; Trp, tryptophan; Tyr, tyrosine; Orn, ornithine; Phe, phenylalanine; Arg. arginine; Kyn, kynurenine.

**Table 4 diagnostics-10-00207-t004:** Kynurenine to tryptophan ratio (KTR) showed a significant association with. ARB responsiveness in diabetic patients with macroalbuminuria after adjusting with other confounding factors.

Models	Multivariate Odds Ratio (95% Confidence Interval)	*p* Value
Unadjusted model	0.639 (0.415-0.983)	0.041 *
Model 1 (age)	0.644 (0.417-0.994)	0.047 *
Model 2 (SBP)	0.619 (0.386-0.991)	0.046 *
Model 3 (eGFR)	0.377 (0.148-0.964)	0.042 *
Model 4 (gender)	0.319 (0.112-0.907)	0.032 *
Model 5 (HbA1c)	0.326 (0.112-0.951)	0.040 *
Model 6 (duration of diabetes)	0.218 (0.057-0.834)	0.026 *
Model 7 (use of DPP4 inhibitor or GLP-1R agonist or SGLT-2 inhibitor)	0.098 (0.012-0.814)	0.032 *

Serum concentration of kynurenine to tryptophan ratio showed a significant association with ARB responsiveness in diabetic patients with macroalbuminuria after adjusting with multiple models of confounding factors by multivariate logistic regression. * *p* < 0.05. Model 1 included the continuous variable of age. Model 2 included the continuous variable of age and SBP. Model 3 included the continuous variable of age, SBP, and eGFR. Model 4 included the categorical variable of gender, and the continuous variables of age, SBP and eGFR. Model 5 included the categorical variable of gender, and the continuous variables of age, SBP, eGFR, and HbA1c.Model 6 included the categorical variable of gender, and the continuous variables of age, SBP, eGFR, HbA1c and duration of diabetes. Model 7 included the categorical variable of gender, and use of DPP4 inhibitor or GLP-1R agonist or SGLT-2 inhibitor and the continuous variables of age, SBP, eGFR, HbA1c, and duration of diabetes. Abbreviations: SBP, systolic blood pressure; eGFR, estimated glomerular filtration rate; HbA1c, glycated hemoglobin A1c; DPP4, dipeptidyl-peptidase-4; GLP-1R, glucagon-like peptide-1 receptor; SGLT-2, sodium glucose co-transporters 2.

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
