# Peer review of "Kynurenine/Tryptophan Ratio Predicts Angiotensin Receptor Blocker Responsiveness in Patients with Diabetic Kidney Disease"

_diagnostics, 2020, doi:10.3390/diagnostics10040207_

Round 1
Reviewer 1 Report
In the present study, the authors included 56 T2DM patients with various stages of chronic kidney disease and micro (mau) or macro (MAU) albuminuria. They measured the level of different metabolites including tryptophan and kynurenine. The authors demonstrated that lower tryptophan level and higher kynurenine/tryptophan ratio (KTR) were significantly associated with MAU. In addition, they found that KTR predicted angiotensin receptor blocker responsiveness in patients with MAU.
The manuscript is well written, and results are convincing. The study, despite a small number of patients, is interesting. My only minor concern is about Orn and Phe levels in mau and MAU patients. All metabolites display the same trend, except Orn and Phe, which are higher in mau patients than in controls, but lower in MAU patients than in mau patients. This should be discussed.
Author Response
Response to Reviewer 1 Comments
Point 1: The manuscript is well written, and results are convincing. The study, despite a small number of patients, is interesting. My only minor concern is about Orn and Phe levels in mau and MAU patients. All metabolites display the same trend, except Orn and Phe, which are higher in mau patients than in controls, but lower in MAU patients than in mau patients. This should be discussed 

Response 1: Thank you for your kind review and generous suggestion. We try to explain our result and will modify our article as below:
1. (Page6 in Results) Out of the 145 metabolites examined, 9 metabolites….showed significant differences among the three groups (Table 2). In contrast to the other metabolites showing a negative association with the severity of albuminuria, the plasma levels of Orn and Phe were higher in the albuminuric, especially in the mau group. In addition, although the plasma levels….
2. (Page 9 in Discussion) To reach this goal, we correlated the change of metabolites in T2DM patients with various stage of CKD in this study, and we found a trend of decreased metabolites (Ser, Trp, Tyr, PC ae C44:3, lysoPC a C24:0, lysoPC a C26:1, SM C26:0) except Orn and Phe, showing significant association with the severity of albuminuria. Several amino acids, including Orn, were reported to be positively correlate with UACR [20], here we also found higher Orn levels in the albuminuric, especially in the mau, rather than in the MAU group of patients. Although we cannot fully explain the discrepancy, one of the possible reasons may be attributed to the use of SGLT-2 inhibitors in the MAU group. Increased ornithine decarboxylase (ODC) activity may lead to hyperfiltration in diabetic kidney [21], any factors interfering eGFR may influence the activity of ODC and further modulate the concentration of Orn. For example in our cases, SGLUT-2 inhibitors may decrease eGFR due to their promotion of glomerular afferent arteriole constriction [22].
Plasma level of Phe is regulated by phenylalanine hydroxylase (PAH), factors such as stress, sepsis or any causes of hepatic dysfunction may influence the activity of PAH [23-24]. In accordance with studies reporting increased plasma Phe level in CKD patients with reduced renal PAH activity [25], we also found higher Phe levels in albuminuric patients, albeit we cannot explain why the Phe level was higher in the mau rather than in the MAU patients.
In corresponding to the previous studies, we also found a significant inverse correlation between the plasma level of Trp…
References: (We will update the references order accordingly in our revised manuscript.)
20. Chuang, W.-H.; Arundhathi, A.; Lu, C.; Chen, C.-C.; Wu, W.-C.; Susanto, H.; Purnomo, J. D. T.; Wang, C.-H., Altered plasma acylcarnitine and amino acid profiles in type 2 diabetic kidney disease. Metabolomics 2016, 12 (6).
21. Thomson, S. C.; Deng, A.; Bao, D.; Satriano, J.; Blantz, R. C.; Vallon, V., Ornithine decarboxylase, kidney size, and the tubular hypothesis of glomerular hyperfiltration in experimental diabetes. J Clin Invest 2001, 107 (2), 217-24.
22. Fioretto, P.; Zambon, A.; Rossato, M.; Busetto, L.; Vettor, R., SGLT2 Inhibitors and the Diabetic Kidney. Diabetes Care 2016, 39 Suppl 2, S165-71.
23. Fuchs, J. E.; Huber, R. G.; von Grafenstein, S.; Wallnoefer, H. G.; Spitzer, G. M.; Fuchs, D.; Liedl, K. R., Dynamic regulation of phenylalanine hydroxylase by simulated redox manipulation. PLoS One 2012, 7 (12), e53005.
24. Morgan, M. Y.; Marshall, A. W.; Milsom, J. P.; Sherlock, S., Plasma amino-acid patterns in liver disease. Gut 1982, 23 (5), 362-70.
25. Saleem, T.; Dahpy, M.; Ezzat, G.; Abdelrahman, G.; Abdel-Aziz, E.; Farghaly, R., The Profile of Plasma Free Amino Acids in Type 2 Diabetes Mellitus with Insulin Resistance: Association with Microalbuminuria and Macroalbuminuria. Appl Biochem Biotechnol 2019, 188 (3), 854-867.
Reviewer 2 Report
Ming-Hsien Wu et al. have reported the association between Kynurenine/Tryptophan ratio and ARB responsiveness in patients with DKD.
â‘ Table 4
Authors should add Incretine therapy (DPP4 inhibitor and GLP-1 agonist) to multivariate binary logistic regression.
Moreover, baseline UACR could influence the alternation of DKD. So, authors sould add baseline UACR to multivariate binary logistic regression.
â‘¡
If authors have information about KTR after treatment of ARB, authors could report KTR after treatment.
Author Response
Response to Reviewer 2 Comments
Point 1: Table 4.
Authors should add Incretine therapy (DPP4 inhibitor and GLP-1 agonist) to multivariate binary logistic regression.
Moreover, baseline UACR could influence the alternation of DKD. So, authors should add baseline UACR to multivariate binary logistic regression.
Response 1:
Thank you for your kind review and generous suggestion. DPP4 inhibitors and GLP-1R agonists may attenuate albuminuria and provide renal protection. Thus we add patients with incretine therapy (4 MAU patients treated with GLP-1R agonist were accidentally omitted in our previous report has been presented in the revised Tab. 1) in model 7 of multivariate binary logistic regression as shown in the revised Table 4.
Furthermore, we agree that baseline UACR may influence the alternation of DKD; nevertheless, when we put the basal UACR into our variables, it became statistically insignificant. We think the substantial variation of UACR (ranging from 330 to 7357 mg/g) in our MAU patients may be the key factor of this result.
Point 2: If authors have information about KTR after treatment of ARB, authors could report KTR after treatment.
Response 2:
We think this point is important and may be beneficial to our study. However, we did not perform second metabolomics study due to some limitations such as budget limitations, patient refusal, etc., thus, only KTR before ARB treatment was surveyed. We will add this point to the limitations in our manuscript.
